# The Measurement of Nanoparticle Concentrations by the Method of Microcavity Mode Broadening Rate

**DOI:** 10.3390/s20205950

**Published:** 2020-10-21

**Authors:** Alexey Ivanov, Kirill Min`kov, Alexey Samoilenko, Gennady Levin

**Affiliations:** 1The All-Russian Research Institute for Optical and Physical Measurements, 119361 Moscow, Russia; asamoylenko@vniiofi.ru (A.S.); levin@vniiofi.ru (G.L.); 2Russian Quantum Center, 143025 Moscow, Russia; k.n.minkov@yandex.ru

**Keywords:** whispering-gallery mode (WGM), microcavity, nanoparticles, concentration sensing, adsorption, measurement system

## Abstract

A measurement system for the detection of a low concentration of nanoparticles based on optical microcavities with whispering-gallery modes (WGMs) is developed and investigated. A novel method based on the WGM broadening allows us to increase the precision of concentration measurements up to 0.005 ppm for nanoparticles of a known size. We describe WGM microcavity manufacturing and quality control methods. The collective interaction process of suspended Ag nanoparticles in a liquid and TiO_2_ in the air with a microcavity surface is studied.

## 1. Introduction

The impact of nanoparticles on humans and the environment requires the provision of various safety measures, sanitary rules and hygiene standards implementation. To assess the risks associated with the production and circulation of nanomaterials, an important task is nanomaterials monitoring; i.e., the detection and quantification of nanoparticles in the environment [1]. According to ecotoxicological studies [2,3], the concentration of Ag nanoparticles at the surface of water bodies reaches values of more than 0.1 μg/L and TiO_2_ of more than 100 μg/L. Studies have also been conducted in wastewater on the surface of water bodies and in weather elements as well as in soil and air.

Silver nanoparticles possess antimicrobial, antiviral and fungicidal properties [4,5,6]. Studies show that silver nanoparticles disrupt the vital functions of a cell by attaching to their surface, which leads to perforation of the cell membrane [7]. Titanium dioxide is a photocatalyst [8]. When exposed to near-UV radiation, TiO_2_ exhibits bactericidal activity [9]. That is why, for safety reasons, it is essential to control the content of nanoparticles in environmental objects.

The requirements for modern nanoparticle detection systems are constantly growing. Measurement speed, the ease of sample preparation and the ability to determine in situ the concentration of the test object in air or water are the main requirements for a modern sensor. Another important requirement is the cheapness of the sensor.

Sensors based on the microcavities with whispering-gallery modes (WGMs) occupy a special place among the test systems [10,11,12]. The advantage of microcavities-based detection methods is their applicability for particles of any type (various shapes, metals and dielectrics). The microcavity, the main element of the measurement system, exhibits high sensitivity. The detection limit for sensors based on microcavities is up to one molecule [13]. WGM microcavities are also used in a number of different types of tasks. A review of microtubular WGM sensors is given in [14]. Such sensors can be relatively inexpensive when mass produced. However, most of the described devices do not have a high optical quality factor (Q-factor), which imposes limitations on the potential sensitivity. Moreover, they require additional technological efforts to feed the analyte into the microtube. Microspheres with Q ~ 10^8^ can be made using simple and reliable technology and can be placed directly into the analyte volume.

The ability to detect nano-objects in various gas and liquid media allows for the use of this sensor to protect the environment and human health from the harmful effects of nanoparticles. WGM sensors are used in the monitoring of dynamic biochemical processes [15]. In addition, the refractive index of the environment [16], temperature [17,18,19], humidity [20], greenhouse gases [21] and deformation [22] can be measured by these kinds of sensors. In addition, the microcavities can be used as a gyroscope [23]. Using microcavities, it is also possible to create stable compact narrow-band laser systems [24,25] and to generate optical frequency combs [26] including soliton ones [27]. These are an important tool for resolving many actual problems; for example, precision spectroscopy [28,29]. It is worth noting that either laser stabilization or generation of frequency combs can be performed using the same microcavity [30,31].

Sensors based on optical microcavities can be used in tasks related to the detection of biomolecules. These sensors demonstrate a sufficiently high sensitivity for clinical diagnostics. Nucleic acids [32], bacterial cells [33], a cancer biomarker [34] and bovine serum albumin [35] were used as biological objects.

The detection mechanisms of nano-objects are based on the interaction of an evanescent field with optical inhomogeneities near the surface of a microcavity. Inhomogeneities on its surface lead to the absorption and scattering of light of WGMs, which changes the microcavity spectrum. Several mechanisms for detecting and analyzing the WGM spectrum are described: resonance frequency shift [36], mode splitting [37], mode broadening [38] and exceptional points [39]. Experiments on nanoparticle detection by mode broadening are described in [40]. This method has a resistance to noise. The simultaneous use of mode broadening and resonance frequency shift mechanisms (i.e., the dissipative and reactive methods), which also gave good results on particle detection, is described in [41]. Recent results in microcavity-based detection demonstrate the ability to detect nanosized particles in aerosols [42].

The main method for detecting particles using a passive microcavity is the resonance frequency shift [43]. The sensitivity of this method is limited by the drift of the pump laser frequency and its amplitude noise. The Q-factor registration method is free from these limitations. In [41] and [44], this method was used for the detection of particles. In these publications, microtoroids made by lithographic technology and stretched fiber as a coupling element were used. The latter, due to its low rigidity, cannot be used in flow systems.

Existing spectrum analysis methods usually provide the possibility to determine either a number of particles deposited on the resonator or changes in the refractive index. For this reason, it is necessary to develop a new method of spectrum analysis for nanoparticle concentration measurement. In this article, we propose the use of the resonance mode broadening rate to measure the concentration of nanoparticles. The proposed algorithm allows us to estimate the concentration of nanoparticles by the rate of their deposition on the resonator surface.

The vast majority of experimental publications describe the detection of nanoparticles in a liquid medium. This is due to the fact that the detection of nanoparticles in aerosols is a more difficult technical task because the calibration of such detectors requires nanoparticle aerosol preparation with specified parameters. A review of recent publications reveals a great interest in the opportunities that microcavities provide in the field of studying the interaction with environmental objects. However, despite the successes in the in-laboratory demonstration of detection methods using an evanescent field, many problems remain in the technical implementation and metrological support of the sensor based on optical microcavities.

The development of a system capable of measuring the concentration of nanoparticles in liquid and gas media is a rather difficult task. The measurement system should be sensitive to low concentrations of particles and have a low cost. For this, the following tasks must be solved: to develop a method and a setup for the production of resonators, to develop methods for measuring the characteristics of microcavities that affect their sensitivity and to create a new algorithm for analyzing changes in the spectrum of a microcavity. The solution of these problems made it possible to create a highly sensitive measurement system for detecting the maximum allowable concentrations of nanoparticles in liquids and gases.

## 2. Materials and Methods

### 2.1. Manufacturing and Preparation of Microcavities

Optical resonators with WGMs are the axially symmetrical bodies made of dielectric material ranging in size from a few millimeters to tens of microns. The advantage of such resonators is their extremely high Q-factor [45]. The geometry and material of microcavities significantly affect their characteristics including the modes of spatial distribution, coupling efficiency, Q-factor and, as a result, sensitivity to external influences. Manufacturing techniques and microcavity shapes may vary significantly. Spherical microcavities are usually made by melting the material with a burner flame or a laser beam [46,47]. There are also techniques for the manufacturing of complex-shaped microcavities using lithography [48] and diamond turning treatment [49]. In addition, microcavities with the shape of microcapillaries have been described [50].

Machining and lithography technologies use complicated and expensive equipment that is not efficient to use in small-scale production. For this reason, we propose to use a microsphere made by melting an optical fiber tip with minimal surface preparation with no selective coating as a disposable (removable) sensitive element for particle detecting. It is interesting to note that the optical fiber itself can act as a sensor for detecting single particles [51].

This type of microcavity with a WGM in the optical frequency range was one of the first to be used and it has been well studied. The refractive index of such microspheres is compatible with the coupling system and the external environment. However, with the manufacturing method by melting, a significant variation in the parameters of microcavities from sample to sample is possible. Therefore, we have developed a unique automated technological setup for small-scale production.

The manufacturing method is based on heating an optical fiber tip in an oxygen-hydrogen mixture flame. To heat the fiber tip, two nozzles are used (Figure 1), positioned so that the flames intersect, balancing the forces of the flame flow acting on the fiber from opposite sides. The preform of fiber is slowly introduced into the cross-flame zone strictly for a certain length (the resonator diameter depends on this). After that, a microsphere is formed at the end of the preform and then it is removed from the heating zone. This allows us to manufacture resonators with high repeatability and symmetry. The advantage of this method is that they are made in one step. Such microspheres are characterized by high reproducibility of diameter and good symmetry (the deviation from the axis of symmetry is less than 5 μm). It should be noted that the shape and symmetry of the microcavity affect the volume of the excited mode. The mode volume, in turn, directly affects the response of the microcavity to particle deposition.

We have developed regimes for the manufacture of microspheres with diameters from 160 to 1800 μm. The root mean square (RMS) of the microresonator diameter of 10% provides a high reproducibility. Such microspheres are cheap to manufacture due to the simplicity of the method and the availability of blanks.

An important task is to control the microspheres’ characteristics. The selective control of microcavities allows us to make adjustments to the technological process of their production if necessary. One of the key parameters of microcavities is the Q-factor. The Q-factor directly affects the particle detection limit. However, the Q-factor control of all microcavities immediately after manufacturing is difficult to implement because Q-factor measuring involves a high probability of its surface degradation with scratches or dust settling. In addition, the measurement of the Q-factor does not provide complete information about the causes of its change.

The Q-factor is mainly determined by the scattering of WGM radiation on various defects. Such defects can be surface roughness and local inhomogeneity of the refractive index of the material. To analyze the quality of the microcavities, we developed a procedure for the microspheres’ selective control by these two parameters.

During microsphere production, inhomogeneities arise due to the melting of the core of the optical fiber. In addition, residual fiber surface contamination can melt into the microcavity and create local refractive index variations in its volume. We researched the influence of the microcavities’ refractive index internal inhomogeneities on their Q-factor by using optical tomography [52]. A standard single mode fiber was used (Model SM600, Thorlabs, Newton, NJ, USA) for the cavity production. The refractive index spatial distribution of the manufactured microcavities is shown in Figure 2b. The results of the Q-factor measurements showed that all manufactured resonators had Q above 10^9^ at the 670 nm wavelength, which was a good result. The difference in the refractive indices of individual inclusions of the core and cladding was 2.5 × 10^−3^. When measuring the Q-factor of a series of microcavities, it was noted that inhomogeneities located close to the surface of the microcavity in its equatorial region degraded its Q-factor. Experiments have shown that the degradation of the Q-factor can be up to three times. It is not possible to determine the Q-factor from a tomogram but it is possible to predict the presence of defects.

As noted above, surface roughness strongly affects the Q-factor of microcavities [53,54,55] and, consequently, affects its detection limit. For a 10 nm size single particle detection, a Q-factor of about 10^9^ is needed [10]. Therefore, the surface quality of the microspheres was tested on a Leica DCM 3D microscope. The non-contact method of confocal microscopy allowed us to measure the parameters of roughness. To collect statistics, we measured the roughness of 20 microspheres in the equatorial region. The measurements were carried out at a base length of 30 μm five times along and across the equator of the cavity. The roughness Ra value was 2 ± 0.5 nm (alongside) and 4 ± 0.5 nm (crosswise).

The Q-factor was determined by the method of measuring the resonant curve width [56]. The measured roughness provided a Q-factor of the microspheres made by the present method from Q = 1.1 × 10^9^ to Q = 3.5 × 10^9^.

Thus, a technological process for the manufacture and control of disposable sensitive elements for a measurement system was developed. The reuse of microspheres is possible if the microcavity is cleaned and the surface is re-prepared. However, it is more expedient and cheaper to make a new one.

The next step for preparing microcavities for operation in a measurement system is surface activation to increase its adsorption properties. This procedure was performed for sensors intended for use in a liquid medium.

The processes associated with the adsorption and particle adhesion on the surface of a microsphere affect the characteristics and sensitivity of such a transducer. Initially, the surface of microcavities is non-selective. From a chemical point of view, the surface of quartz (SiO_2_) is similar to the surface of dispersed amorphous silica. Oxygen-containing compounds located on the surface of silicon, in most cases, are in the form of two reactive groups—silanol and siloxane. If it is necessary to detect elements that do not interact with siloxane bonds, the assembly method on a surface should be used. This multi-stage process requires a well-established technique, is complicated from the technological point of view and also allows it to detect only strictly defined types of substances [57]. However, it has an advantage. A microcavity with a selective coating allows for the detection of particles of a known chemical. It is worth noting that the quality factor will decrease after the application of a selective coating, which can sometimes be critical for highly sensitive measurements of single nanoparticle interactions.

Particles on the silica surface are fixed due to the opening of siloxane bonds and the formation of two silanol groups. This can be achieved in several ways: by exposure to temperature and solutions of mineral acids, UV-ozonation [58] or exposure to distilled water.

Hydroxylation of the surface of quartz using exposure to distilled water has several significant advantages. It does not require local heating to several thousand degrees and it does not reduce the quality factor due to the film that is formed in the solution of mineral acids. We experimentally chose the time of hydroxylation in distilled water at room temperature, sufficient to observe stable adsorption of particles in a liquid. The hydroxylation procedure consisted of holding microspheres in distilled water for one hour while the siloxane bonds were broken with the formation of silanol groups bound to the surface by quartz molecules and having high reactivity. Particle adsorption on a microcavity was experimentally verified using dark field microscopy [59]. It is worth noting that if the microsphere after the hydroxylation procedure is exposed to air, then the siloxane bonds are closed and the microcavity loses its previous adsorption properties.

### 2.2. Measurement System

After preparing the microcavity surface, it was placed in a cuvette of the measurement system. The task of the system was to record the spectral characteristics of the resonators in dynamics to detect the broadening, shift or splitting of the WGM resonance peaks. Its scheme is shown in Figure 3.

A tunable diode laser with an external resonator ECDL6707R at an average wavelength of 670 nm was used as a coherent light source. The laser continuously scanned the microcavity in frequency. The beam fell on BS1 in such a way that part of the light was sent to the measuring cuvette and part to the reference interferometer. The reference ring interferometer made it possible to perform frequency calibration determining the loaded Q-factor of the microcavities.

A microcavity with a coupling system was located in the cuvette. The coupling system was a prism that coupled a laser light with a microcavity (under-coupling regime) and was used for the input/output of radiation. The under-coupling regime was determined experimentally during tuning by the absence of a change in the width of the resonance curve with small variations between the coupling elements and the microcavity. We used the estimation of the coupling coefficient as reported in [60,61]. In the cuvette, the microcavity had four degrees of freedom, which were three translational and one rotary around its own axis. In the vicinity of the microsphere, a camera with a ×10 lens was installed to monitor its position and WGM excitation. The radiation passed through the microcavity along with the radiation reflected from the edge of the prism and fell on the photodetector where the intensity of the incident light was converted into voltage. Thus, the transmission spectrum of the cavity with a negative peak whose width Δν characterized its loaded Q-factor was visible on the oscilloscope.

The system for feeding the sample into the cuvette varied for different channels. A peristaltic pump was used to feed liquid analyte to the cuvette. The use of a peristaltic pump was dictated primarily by the need for no contact of the pump mechanism with the liquid sample while the dosage of the liquid occurred with high accuracy.

An air compressor was used to measure aerosols in the second channel. When calibrating the air channel, a nebulizer and dehumidifiers were additionally connected to the measurement system (shown in Figure 3). The procedure for producing particles suspended in air prepared from a hydrosol is described in detail in [62].

### 2.3. Measurement Method and Processing Algorithm

An important component of the sensor is the signal processing algorithm because it determines the measuring algorithm and its characteristics.

The method of detection using the mode broadening rate is based on the adsorption of nanoparticles on the microsphere surface and their interaction with the evanescent WGM field. As a result, the transmission spectrum of the microcavity changes in time. At the start, immediately after the nanoparticles are fed into the measuring cuvette, particles located near the adsorption centers on the microsphere surface have a chance to become bound. Each adsorbed particle reduces the loaded Q-factor of a microcavity. After some time, most centers will be filled with particles and the surface of the microsphere will be “filled”. As a result, to estimate the particle concentration, one has to analyze the Q-factor degradation curve where time is plotted along the *x*-axis and WGM resonance width along the *y*-axis. The initial linear portion of the curve is the most informative. It contains information about the particle adsorption rate, which is proportional to the number of particles near the microcavity surface WGM region. The slope of this section is the main measured parameter of the system.

The workflow scheme of the measuring signal processing algorithm is shown in Figure 4. Before the measurement, one mode was selected from the resonator spectrum and then monitored. Signals were recorded from the photodetectors PD1, PD2 and the laser control unit (Figure 4a). At the first stage of signal processing, the transmission signal of the microcavity was divided into a sequence of individual spectra. After that, a spectrogram for monitoring the observed mode was constructed from these spectra. The spectrogram (Figure 4c) is a matrix where the rows represent the laser frequency detuning and the numbers of the spectrum as it changes in time are plotted vertically. The signal intensity is shown in color.

The initial spectrogram (Figure 4c) has a zigzag structure because of the resonant frequency drift due to the temperature change and adsorption of nanoparticles. In this method, the resonance shift has no useful information and must be filtered out. Therefore, a straightened and normalized spectrogram was calculated, which clearly demonstrated the mode broadening. For this purpose, a linear slope was subtracted at each time for each implementation of the spectrum, which was caused by a change in power when the laser frequency was adjusted. After that, the spectrum was normalized so that the peak value of the spectrum reached a value of 1 and the opposite value at a minimum of 0. The resonant frequency of the peak was then calculated using the method of “center of mass”. Further in the spectrogram, for each implementation of the spectrum, its center of mass was shifted to a central position.

At the next stage of processing, using the threshold filter, the peak width Δν (Figure 4d) at half maximum was extracted. As a result, the Q-factor degradation curve was obtained as a function of time (Figure 4e). The final stage of the algorithm’s operation was the fitting of this curve with the adsorption model and subsequent determination of its slope at the initial point, which gave the quality factor change rate. It is also possible to fit a non-linear curve by the Langmuir equation [63] or simpler, which gives good accordance [64].

A limitation of this algorithm operation is the situation when, at high concentrations, the formation of a “filled” film of nanoparticles on the surface of a microcavity occurs in a time comparable with the laser scanning frequency (in this case, *f* = 25 Hz). As the particle deposition is a dynamic process, it is very important to accurately determine the initial moment of time when the colloid is fed into the measuring cuvette.

## 3. Results

### 3.1. Graduation and Experiment

A series of experiments was carried out to measure the concentration of nanoparticles in a liquid and air medium. Silver nanoparticles were used as a sample for a liquid medium and titanium dioxide for an air medium. These nanoparticles are widely used in industry and medicine. Spherical silver nanoparticles were produced by the Leopold and Lendl method [65]. As a standard solution for measurement system calibration, a colloidal solution with spherical nanoparticles 10 nm and 100 nm (average diameter) stabilized with polyvinylpyrrolidone (PVP) and diluted in distilled water was used. A colloid with spherical particles of 50 nm diluted in ethylene glycol was also used. Colloidal solutions were diluted to obtain the following mass concentrations: 0.005, 0.05 and 0.5 ppm for particles with a diameter of 10 and 100 nm and 0.05, 0.5 and 5 ppm for particles with a diameter of 50 nm. The results for the liquid medium are shown in Figure 5. A logarithmic model was used to approximate calibration graphs. During measurements, a colloidal solution was fed to the cuvette and a change in the quality factor was recorded. For the majority of samples, all adsorption centers of the microcavity became filled and the subsequent degradation of the Q-factor did not occur within 8–10 min. During this time, the width of the resonant peak increased by 10–15 times.

These graphs show that different mode broadening rates corresponded to different sizes of nanoparticles. This effect in a liquid can be explained by the diffusion coefficient dependence on particle size, which is determined by the Stokes–Einstein relation. Thus, to calibrate the sensor in a liquid, it is necessary to take into account the mean particle diameter and temperature. It is also worth noting that, by knowing the concentration of the colloidal solution, it is possible to estimate the particle diameter from the calibration graphs. From this experimental data, we concluded that the rate of the Q-factor degradation of microcavities increased proportionally to the logarithm of the nanoparticles’ concentration in a liquid medium.

In a gas medium, the system was calibrated with TiO_2_ particles. The titanium dioxide nanoparticles’ aerosol was prepared in three mass concentrations: 0.001 mg/mL, 0.03 mg/mL and 1 mg/mL in accordance with the procedure [66]. Detection in the air was carried out as follows. The compressor supplied airflow at a rate of 2 L/min to a nebulizer flask filled with a hydrosol. At the output from the nebulizer, the aerosol made from hydrosol containing nanoparticles was obtained. Passing through the dehumidifiers, the water evaporated and, as a result, an aerosol of titanium dioxide nanoparticles fed into the cuvette. At the output of the dehumidifier, the integral number concentration of the aerosol was 1.5 × 10^5^, 2.9 × 10^5^ and 7.9 × 10^5^ pcs/cm^3^, respectively, which was confirmed by measurements using the differential electrical mobility method. The effective size of the particles was in the range from 3 to 35 nm.

The calibration results did not allow a quantitative measurement of the concentration of nanoparticles by the mode broadening method. Despite the fact that the shape of the Q-factor degradation curve was of the same nature as in experiments in a liquid medium, gradual filling of adsorption centers and reaching filling, the mode broadening rate did not increase with a change in the number concentration of nanoparticles in the cuvette. A probable reason for this result was the coagulation of particles in the cuvette and pipelines of the measurement system. Coagulation (the process of particles sticking together when they collide) is one of the main factors in the evolution of disperse systems. The dispersed system tends to reduce free surface energy by reducing the specific surface of the particles. Thus, when sticking together, the particle size changes due to aggregate formation, which significantly affects the aerodynamic characteristics of the particles. At the same time, the adsorption and dynamic properties of such particles change. In turn, it leads to a change in the deposition rate and the corresponding error in the measurements by the considered method.

However, the system exhibited a clear response to the presence of the particles in the cuvette. The operation of the air channel in this case was possible in the qualitative mode. However, it is possible to detect the events of individual deposition of large particles by the mode broadening curve.

### 3.2. Error Estimation

Early studies have shown that the sensitivity of the sensor depends on the size of the resonator and the mode volume [67,68]. We also experimentally investigated the effect of the microcavity mode volume on the measuring signal and the system sensitivity. As a new microcavity was used for each measurement, the mode volume for each microsphere was different. The mode volume may strongly influence the scatter of the measurement results. This fact was revealed in a series of experiments on the nanoparticle deposition on the microspheres. Microimaging was used to estimate the mode volume in the measurement system. Figure 6 shows microphotographs of microcavities with an excited WGM where the luminous mode belt is visible. The interaction area can be estimated from the belt width.

For each particle adsorbed on the surface of the microsphere, the relative contribution to the light scattering of the mode will depend on the spatial localization of the mode. For a mode with a smaller volume, each particle accounts for a significant part of the energy loss of this mode relative to the mode with the wider belt. This is due to the difference in the spatial area of interaction. This effect is observed experimentally in a liquid and air medium and to a large extent determines the sensitivity of the sensor. The observed width of the WGM belt was in the range from 50 to 140 μm, which for the same diameter of the microcavity corresponded to the active interaction area from 0.04 to 0.13 mm^2^. System tests have shown that when reproducing the same concentration of nanoparticles in a cuvette, the results can vary significantly. In Figure 6, we can see that with an increase of the active interaction area, the sensitivity of the method decreased. This approximation curve type is explained by the fact that the mode broadening was proportional to the square of the modes’ coupling coefficient, which depended on the mode volume [38]. This is why it is important to consider this parameter in order to reduce measurement errors.

We also analyzed the main sources of error in the optical design of the sensor [69], which can affect the measurement of the resonance peak width. These sources are listed in Table 1. The error values are given both as the peak width (MHz) and the mode broadening rate (MHz/s).

The experimental results of the measurement system tests show acceptable accuracy in a liquid medium. When using a standard solution of a colloid of Ag particles with sizes of 10 nm and 50 nm, the spread in the concentration values in the experiments did not exceed ±0.05 ppm over the entire concentration range under study. However, when using particles with an average diameter of 100 nm, the calibration characteristic became shallow and the measurement error increased. One of the reasons we were not able to calibrate the system for different concentrations could be due to a high error margin. This fact also limits the application of the method for large particles. It is worth noting that the measurement error of the mass concentration was calculated from the calibration curve for each specific type of medium, nanoparticle and measurement condition. The sedimentation kinematics were different for each case.

## 4. Discussion

Experiments conducted on microcavities without selective coatings showed the possibility of using a sensitive element with minimal surface preparation. Nevertheless, these results allow us to further develop the principles of detection using the mode broadening rate method with the use of the modified microcavities’ surface.

In general, the error of the measurement system was at the level of shop instruments for measuring the parameters of disperse systems. The error of the method, expressed in units of concentration (mg/L or ppm), depends on the slope of the calibration curve and, in some cases, this method can only be used as an indicator. In a liquid, the detection limit of the method was a few particles with a diameter of 10 nm. Therefore, it is possible to obtain a response from a WGM microcavity if the threshold of maximum permissible concentrations is exceeded.

To further improve the accuracy of the method under consideration, it is necessary to provide several preparatory procedures and mandatory measurement requirements:To control the active area of interaction and the diameter of the microcavity. Use microcavities with a small mode index and a WGM belt width of not more than 50 μm (with a microsphere diameter of 300 μm) to measure nanoparticle concentrations below 0.01 ppm;To use individual containers for storing microcavities. Protect the surface of the microcavity from accidental contact and pollution. Place microspheres in sealed containers immediately after manufacturing;Provide vibration protection. Use rigid clamping of the microcavity and coupling elements relative to each other;Use a thermally stabilized measuring cell with temperature control inside the cuvette;Use a tunable laser with lower amplitude and frequency noise (less than ±0.45 MHz). Work in the scanning range away from the region of the laser mode hopping where the spectral quality of the laser radiation varies slightly;To calibrate the system with particles with a known average size in a medium with a known viscosity coefficient and refractive index.

## 5. Conclusions

A new method and an algorithm for nanoparticle detection using microcavities has been developed. This method can be applied to a liquid medium. Experimental results show the ability of the proposed system to measure with high precision the concentration of nanoparticles in a liquid medium (of the order of 0.005 ppm) and to detect the presence of particles in the air (concentration above 0.001 mg/mL). We have developed a simple and cost-effective technology for the fabrication of microspheres by heating an optical fiber tip from two sides. It allows one to create microcavities with reproducible characteristics of geometry and a quality factor of the order of 10^9^. Proposed methods for the real-time monitoring of the microcavity transmission spectrum enable the development of a reliable sensor for nanoparticle detection. Hydroxylation of the surface of the microsphere makes stable adsorption and the detection of particles in a liquid possible.

The mode broadening rate method experimentally demonstrates that the rate of the microcavities’ Q-factor degradation increases proportionally to the logarithm of the nanoparticles’ concentration in a liquid medium. The effect of a decrease of the Q-factor degradation rate with an increase in the radius of nanoparticles is also observed.

The approaches used in this work reduce the cost of measurement systems using microcavities as disposable interchangeable sensing elements. This will allow for the application of this method to air and water pollution control of nanoindustry wastes. The advantages of this type of sensor lead to the effective and inexpensive implementation of the detection of various nanoparticles. Furthermore, the possibility of modifying the surface of microcavities allows one to expand the range of detectable substances and use this sensor not only for the detection of nanoparticles but also for biodetection; for example, the detection of cancer cells, virions, antigens, DNA molecules, RNA and proteins.

## 6. Patents

As part of this work, a setup for the production of microcavities was patented. RU 2,700,129 C1.

## Figures and Tables

**Figure 1 sensors-20-05950-f001:**
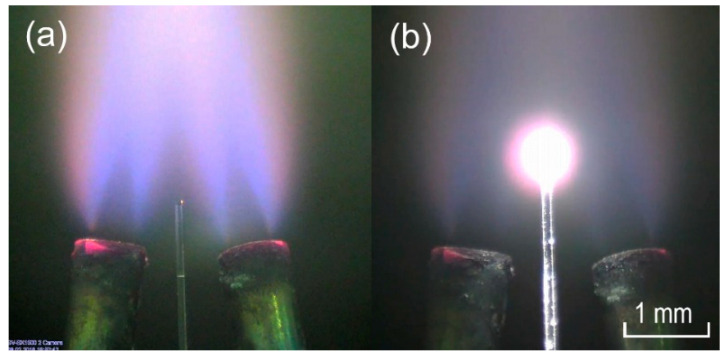
The process of forming a microsphere from an optical fiber in an oxygen-hydrogen burner flame: (**a**) The moment when the preform is introduced into the flame; (**b**) The formation of a microcavity.

**Figure 2 sensors-20-05950-f002:**
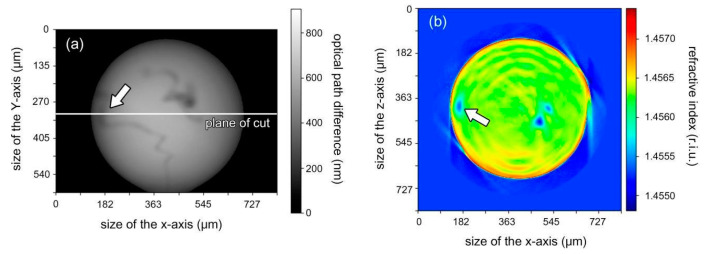
The results of the microcavity local inhomogeneities study made by optical tomography [52]: (**a**) Phase image of a microcavity, side view; (**b**) Two-dimensional tomogram slice of a microcavity in the equator area. The color shows the spatial distribution of the refractive index. The arrow indicates the local region where the defect is located, which appeared as a result of the optical fiber core melting.

**Figure 3 sensors-20-05950-f003:**
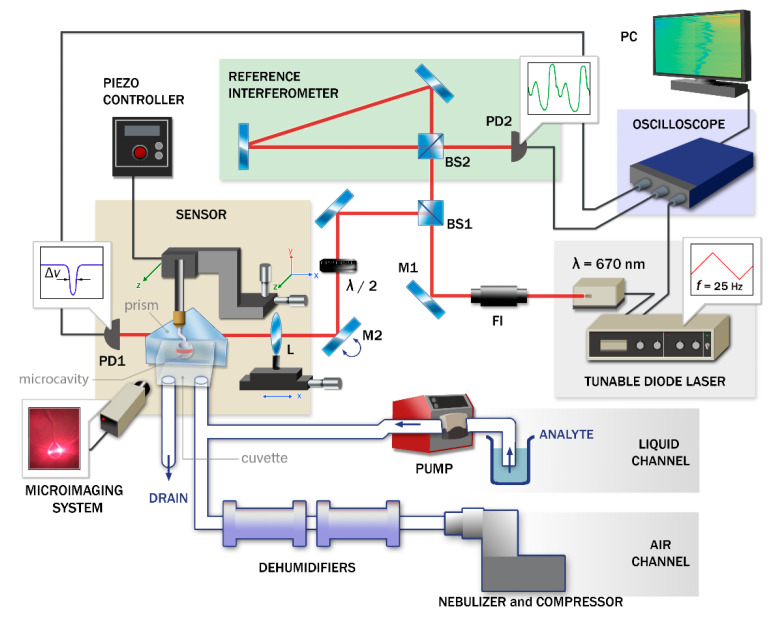
A scheme of a two-channel measurement system for liquid and gas media: PD1, PD2—photodetector; BS1, BS2—beam splitter; L—lens is mounted on a micrometric slide; M2—kinematic mounts for round optics with mirror; FI—free-space isolator; λ/2—half-wave plate.

**Figure 4 sensors-20-05950-f004:**
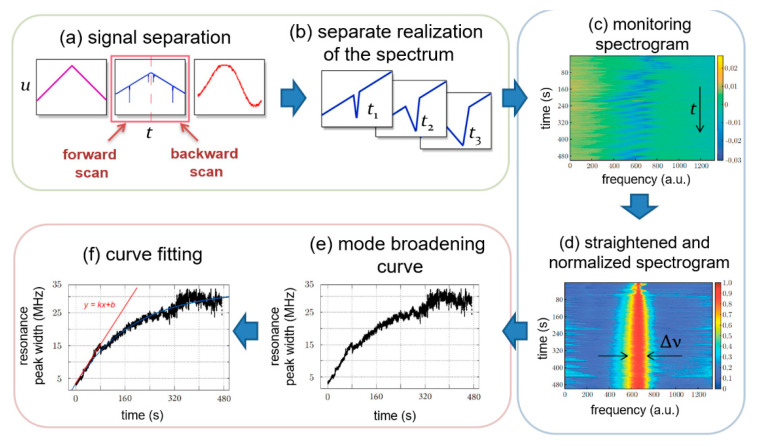
Measuring signal processing algorithm: (**a**) Signals from the laser control unit (violet), microcavity (blue) and reference interferometer (red); (**b**) Separation of the signals containing resonant peaks from the microcavity into individual spectral realizations in time; (**c**) Visualization of a spectrogram of mode monitoring; (**d**) Normalization of the spectrogram and compensation of the resonant frequency shift; (**e**) Selecting a resonant peak and plotting its width at half maximum at each moment in time; (**f**) Approximation of the mode broadening curve and finding the slope of its initial section.

**Figure 5 sensors-20-05950-f005:**
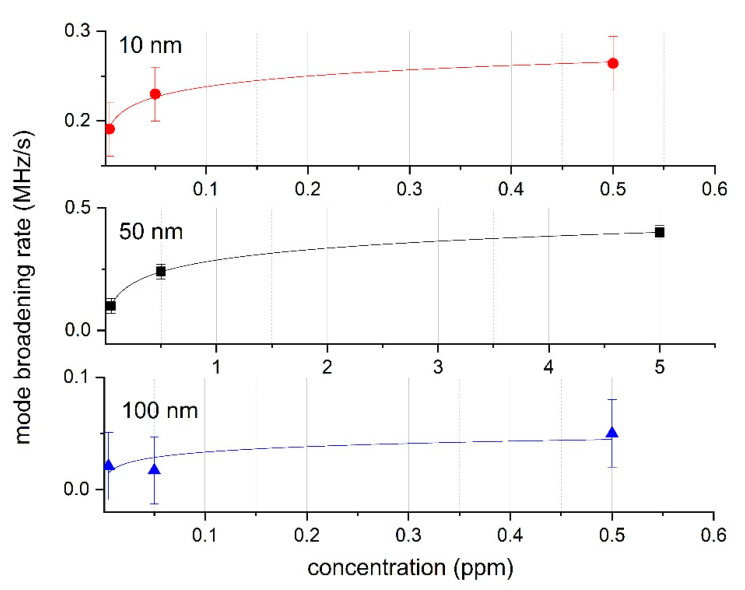
A calibration graph of the dependence of the mode broadening rate on the concentration of nanoparticles in a colloidal solution for particles with different sizes: 10 nm (red), 50 nm (black), 100 nm (blue).

**Figure 6 sensors-20-05950-f006:**
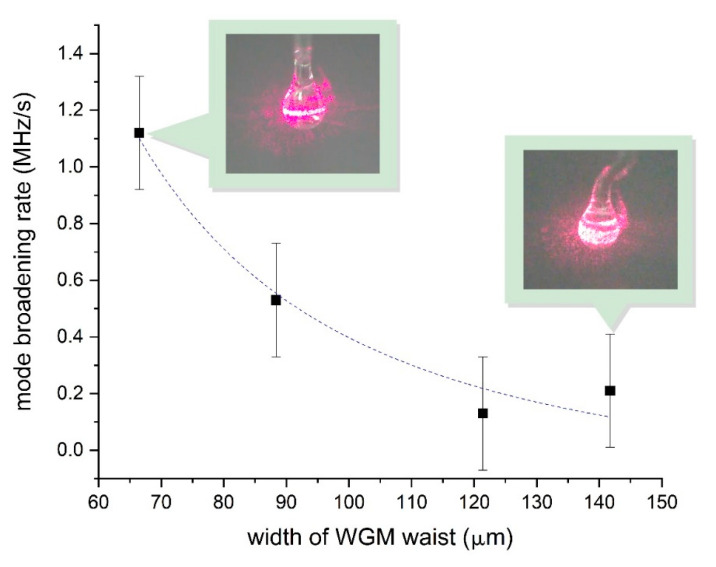
The dependence of the measurement system sensitivity on the width of the WGM belt. The dashed line shows an inverse-square fit. The insets show microcavities with excited WGMs and adsorbed particles with different active interaction areas.

**Table 1 sensors-20-05950-t001:** Additional sources of error.

Factors Affecting Error	Peak Width Value	Broadening Rate Value
Reference interferometer	±0.65 MHz	±0.03 MHz/s
Total instability of tunable laser [70]	±0.45 MHz	±0.02 MHz/s

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
