# Peer review of "The Measurement of Nanoparticle Concentrations by the Method of Microcavity Mode Broadening Rate"

_sensors, 2020, doi:10.3390/s20205950_

Round 1
Reviewer 1 Report
This work is about new methods to detect and also, fabricate nanospheres.
Although the authors claim that these are novel methods, I would like to see more references on past work in the same specific areas and outline clearly how this work differ from what is already in the literature. In addition, the authors refer to quote, 'Machining and lithography technologies use complicated and expensive equipment that is not efficient to use in small-scale production': this statement is not true as there are competitive fabrication technologies nowadays that are less expensive such as nano imprint as an example that may be suitable and may offer more reproducibility than the fabrication method described here.
In addition, I do find that too much details on the set-up and experiment in body of content are provided in this paper. The content appears more suitable for an appendix as some of them are routine processes that are not necessarily novel. As a result, they may be shifted in other section as a supplement.
Reviewer 2 Report
In this paper, the authors report a novel scheme of monitoring the nanoparticle concentrations in both air and liquid environment using ultra-high-Q whispering gallery mode optical microcavities. The characterized Q factor of fabricated microspheres (above 10^9) is very impressive. The slope extraction of mode broadening is technically sound. The measurement setup is carefully designed with a novel sample delivery configuration. The algorithm looks reasonable and efficient to extract mode broadening information and minimize the noises. In particular, the investigations on the waist of cavities through the scattering images are very interesting. Moreover, sufficient technical details and discussions were provided.
Therefore, I would recommend this paper for acceptance, and also ask for the attention of addressing the following points in the minor revision.
- Adding a scale bar in Figure 1 will be useful.
- In Figure 3, an optical microscope imaging showing the microsphere, the prism and the integrated fluidic chamber will be helpful.
- Is it possible to calibrate the spectral sensitivity here (meaning the spectral shift as a function of bulk refractive index change)? Then this figure-of-merit can be fairly benchmarked with those in previously demonstrated whispering gallery microsensors.
- On Page 6, the authors state “The coupling system is a prism that couples a laser light with a microcavity (under-coupling regime) and is used for input/output of radiation. ” How was the under-coupling condition determined here? Any estimation of the coupling coefficient? Besides, presenting a transmission spectrum of resonant modes will be very useful.
- In Figure 5, the standard deviations of data for 10nm-sized particles and 100nm-sized particles are very high. Besides, the data for 100nm-sized particles even suggest a decreasing trend for the first two data points. The author should at least comment on theses.
- The authors have made very nice overviews on previous nanoparticle sensors made by solid-core whispering gallery mode microcavities, such as microtoroids and microspheres. There are some other whispering gallery mode optical microsensors such as hollow-core microtubes (e.g., see 1021/acssensors.9b00681) showing the capability of monitoring molecular dynamics at the sensor surface such as water molecules, dye molecules. For some of the work, monitoring of the Q factor is also adopted. Microtubular-shaped optical sensors may suggest a decreased Q factor but increased sensitivity due to the strong evanescent fields. The authors could also comment on the “pros and cons” of this scheme compared to hollow-core whispering gallery mode optical cavities for nanoparticle detection.
Reviewer 3 Report
Overall the quality of writing can be improved. For example, the symbol of liter should be capital L. Additionally, "giant" quality factor should be replaced with "extremely high quality factor". Introduction section should be improved as it is mentioned that there is requirement for low cost measurement setup but figure 3 in manuscript shows a very elaborate measurement setup. The word "measuring system" should be replaced with either "measurement setup" or "measurement system". All symbols in Figure 3 should be defined. For example, M1, BS1, L1, PD1 etc. These symbols will not be known to non-expert readers. Figure 5 shows that results from 50 nm particles are very well controlled (low standard deviation/error). One of the reasons author are not able to calibrate the system (Page 9, lines 318) for different concntrations could be due to high error margin. You can see Figure 5 and it clearly shows that error margins overlap with different particle concentrations (10nm and 100 nm graphs). This should be included in the manuscript.
Author Response
Please see the attachment.

This manuscript is a resubmission of an earlier submission. The following is a list of the peer review reports and author responses from that submission.
Round 1
Reviewer 1 Report
The authors present a method to estimate nanoparticles concentration in a liquid by means of the broadening rate of microsphere WGM resonance-bandwidth. The authors exploit a linear fit of the initial time portion of the Q-factor degradation curve for extrapolate information about particle adsorption rate which is proportional to the number of particles near resonator surface.
The manuscript is a cut-and-paste of previous contributions of the same authors:
_Section 2.1: Patent RU 2700129 C1.
_Section 2.3: REF [53]: Ruzhitskaya et al. “Analysis of the Transmission Spectra of Optical Microcavities Using the Mode Broadening Method”; Optoelectronics, Instrumentation and Data Processing. 2018, 54, 8;
_Section 3.2: REF[56]: Ivanov PhD Thesis “Measuring system based on optical microresonators for detecting small concentrations of nanoparticles”.
The incremental contribution is really poor and the original sections are not well-developed.
Last but not least, in section 3.2 there are important sentences about sensitivity which are not supported by robust arguments and sufficient experimental evidences (fig.7).
Reviewer 2 Report
The authors report on a method of measuring nanoparticle concentration based on the rate of mode broadening in a whispering-gallery microresonator. They describe a new method for fabrication of silica microspheres and their characterization by optical tomography. They develop a signal processing algorithm that takes the transmission signal power resonance dips and converts them to a spectrogram, from which a mode broadening curve is extracted. The initial slope of the broadening curve then gives the nanoparticle concentration. This method is then experimentally demonstrated, quantitatively using silver nanoparticles in liquid, and qualitatively using titanium dioxide nanoparticles in air.
The paper contains worthwhile contributions: the fabrication and characterization techniques are novel and interesting, and the signal processing algorithm is an improved way of monitoring change in linewidth. However, the description of methods and presentation of results are not up to publication standards; for example, the title is an example of the extensive English editing that is needed. In my opinion, the expansive revisions that are needed go beyond simply major revisions, and I recommend that the manuscript not be accepted for publication in its present form. If the authors wish to make substantive changes and submit a new paper for consideration, a few points for them to address are given here.
(1) In line 105, the low thermal expansion coefficient of silica is mentioned as significantly reducing thermal drift. However, the thermo-optic effect (change of refractive index with temperature) is ten times larger than that of thermal expansion in causing mode shifts with temperature changes.
(2) The fabrication method shown in Figure 1 and described in the accompanying text should be described in more detail. For example, from the figure, it looks like the orientation of the flames is changed during the process – is that the case?
(3) In line 142, it is said that the Q of the microcavity can be determined from the tomogram shown in Figure 2. How is this done?
(4) In the fabrication process, it is not specified whether the fiber that is used is single-mode fiber.
(5) The roughness parameter Ra shown in Table 1 is not defined, and the measurement process is not described in sufficient detail.
(6) The determination of Q from the results in Table 1 is not specified. Was the method of Ref. 47 used?
(7) In lines 184-185, it is mentioned that hydroxylation does not reduce the Q due to formation of a surface film, but nothing is said there or later about the fact that the Q actually used (determined from Fig. 4) is two orders of magnitude smaller than the presumed Q of line 162.
(8) In Fig. 3, it is clear that the transmission spectrum (in the sense used in Ref. 34) is being measured, so “reflection” in line 211 must be a misprint. “Transmission” is used later, in lines 227 and 239.
(9) It seems to me that the crucial part of their signal processing algorithm in Fig. 4 is the step from (c) to (d), especially the normalization of the spectrogram. This should be described in more detail.
(10) In Eq. (1), mode broadening in the reflected spectrum is not needed. In these expressions from Ref. 34, only the backscattering contribution to mode broadening is considered. It was shown in Ref. 34 that for 10 nm silver nanoparticles at nearly the same wavelength, the contribution to linewidth broadening from nanoparticle absorption is about ten times larger than the scattering contribution. This makes a comparison between model and experiment suspect.
(11) The parameters listed in lines 276-279 should be described in more detail, especially g.
(12) In line 285, the units of g are not given, and the expression for its randomized value is confusing.
(13) In line 286, the value given for the mode volume is more than two orders of magnitude larger than the volume of the entire microsphere, so there is obviously an error.
(14) Figures 5 and 6 are very poor presentations of the model and results. A standard 2D plot should be sufficient for each, and much easier to read. The horizontal axes need to be labeled, and the logarithmic scale mentioned in lines 311-312 clarified. Based on Eq. (1), the broadening is proportional to N, not to some other function of N.
(15) In Fig. 7, why was a linear fit presented? The broadening is proportional to the square of g, and g is inversely proportional to the mode volume (as seen in Ref. 34), so it would have been better to have an inverse-square fit.
(16) Finally, in the References, Refs. 34 and 37 are cited incorrectly.
Reviewer 3 Report
The manuscript by Ivanov et al demonstrates the detection of the low concentration nanoparticles using mode broadening speed of a whispering gallery mode microsphere. The idea is interesting and the experimental results are solid, which could be accepted after some minor revisions. I have some comments and suggestions listed below.
1. As shown in Fig. 4f, the slope of the mode broadening is a nonlinear curve instead of a linear one. In this case, it will be hard to fit a linear slope. Is it possible to fit a nonlinear curve and using the duration time of mode broadening as a signal, please refer Optics express 26 (1), 51-62 (2018).
2. It seems the device (microsphere) could be used only once. Is it possible to clean and reuse the sphere after measurement?
3. What is the detection limit of the current sensing mechanism?
4. Some related papers are missing in the reference list, such as arXiv:1805.00062; Light: Science & Applications 9, 24 (2020); Advanced Materials 26 (44), 7462-7467 (2014).